# "Two shots for life. HPV-vaccination at your school". Co-creation of a complex intervention to reduce ethnic inequity in childhood HPV-vaccination in Denmark

Anne Katrine Leonhard[1,2]*, Sara Koed Badre-Esfahani[1], Mette Bach Larsen[2,3], Lone Kjeld Petersen[4,5], Lene Seibæk[1,2]

1 Department of Gynecology and Obstetrics, Aarhus University Hospital, Aarhus, Denmark, 2 Department of Clinical Medicine, Aarhus University, Aarhus, Denmark, 3 Department of Clinical Medicine, Horsens Regional Hospital, Horsens, Denmark, 4 Department of Gynecology and Obstetrics, Odense University Hospital, Odense, Denmark, 5 Department of Clinical Research University of Southern Denmark, Odense, Denmark

☉ These authors contributed equally to this work.

* akle@clin.au.dk

## Abstract

### Background

Despite free access to HPV-vaccination, adolescents with an ethnic minority background have an overall lower vaccination coverage due to various barriers needed to be accommodated in a targeted effort. We aim to report the development process of an intervention to increase HPV-vaccination coverage among adolescents with an ethnic minority background.

### Methods

An iterative co-creation guided by Medical Research Council UK development of complex interventions framework was done. We involved stakeholders throughout the process from the analysis of intervention interest and impact. The development process is described in four phases: 1) *Evidence synthesis* with focus-group interviews and literature review 2) *Acceptability* with stakeholders, 3) *Elements and context* with creation of a logic model, and finally 4) *Co-production* of material.

### Results

The final intervention, "Two shots for life", is a school-based intervention consisting of HPV-education targeting adolescents and parents and HPV-vaccination at school during school hours. The intervention was designed to require the fewest possible resources for implementation sites, being municipality schools with a proportion of pupils >25% with ethnic minority background. Communication focus is

**Data availability statement:** All relevant data are within the paper and its Supporting Information files.

**Funding:** This study was funded by Danske Regioner (https://www.regioner.dk/, grant EMN-2019-00852 1431576 received by SBE), Region Midtjylland Folkesundhed i Midten (https://www.sundhed.rm.dk/, grant R72-A3616 received by AL), Region Midtjylland Folkesundhed i Midten (https://www.sundhed.rm.dk/, grant A3616 received by AL), Region Midtjylland Sundhedsvidenskabelige Forskningsfond (https://www.sundhed.rm.dk/, grant A4149 received by AL), Familien Hede Nielsens Fond (https://www.hedenielsens-fond.dk/, grant 1492 received by AL) and Helsefonden (https://helsefonden.dk/, grant 22-B-0247 received by AL). The funders had no role in study design, data collection and analysis, decision to publish, or preparation of the manuscript.

**Competing interests:** All authors declare no competing financial interests or personal relationships that could appear to have influenced the scientific work reported in this paper.

HPV-vaccination as a prevention effort, it is mainly delivered face-to-face and written material is available in relevant minority languages as well as visual in appearance.

## Conclusions

A combined intervention with both educational and logistical components created with stakeholders is expected to accommodate barriers for HPV-vaccination in ethnic minority communities and increase accessibility and, hence, coverage of HPV-vaccination. By presenting a thorough development process, we expect to achieve transparency that will support future development of similar complex interventions aiming to increase HPV-vaccination coverage among ethnic minorities.

## Trial registration

ClinicalTrials.gov NCT05681169.

---

## Introduction

Human Papillomavirus (HPV)-related cancers are considered the most preventable forms of cancer due to effective vaccination against HPV [1–4]. Since its introduction in 2006, organized HPV-vaccination, alongside cervical cancer screening, has contributed to the reduction of cervical cancer incidence and mortality in many western countries [5,6]. Besides cervical cancer, HPV-related cancers also cover penile, vulva, vaginal, anal, and oro-pharyngeal cancer. Both anal and oro-pharyngeal cancers are increasing in incidence [7–9].

Despite free access to HPV-vaccination it is a tendency in Denmark and other western countries, that citizens with ethnic minority background have an overall lower HPV-vaccination coverage [10–15]. Barriers for ethnic minority adolescents regarding HPV-vaccination are complex and includes insufficient awareness among parents, partly due to inaccessible information. Adding to this, cultural taboos related to sexuality, emotional barriers such as fear that vaccination could lead to promiscuity, unawareness of prevention strategies, and a general mistrust in general practitioners (GPs) are of importance [16–19].

School-based HPV-vaccination has shown to improve coverage among children of all ethnicities [20], possibly because of its convenience and peer support at time of vaccination leading to self-efficacy [21,22]. In addition, a high trust in school systems mitigates some of the barriers against HPV-vaccination. Studies have shown positive attitudes toward HPV-education supplementing school-based vaccination [23,24], due to insufficient understanding being a main barrier for vaccination regardless of ethnicity [25–28]. However, to successfully increase attendance of ethnic minority children, it is emphasized that interventions need to be culturally adapted, multi-phased, and targeted communication must follow guidelines regarding communication with ethnic minority citizens about health issues [29–32]. Thus, despite well-established knowledge regarding barriers for HPV-vaccination applicable to ethnic minority children, HPV-vaccination programmes, and HPV-educational packages,

no study has yet addressed these elements in a combined targeted intervention with the aim of increasing vaccination-coverage for children across ethnicities.

The aim of this paper is to describe the development process of such a complex intervention combining these elements to increase HPV-vaccination coverage among adolescents with ethnic minority background. We will start off by describing our theoretical framework and progress of intervention design then give an overview of the final intervention and its content.

## Method

The intervention was considered complex from the outset, so development followed the Medical Research Council (MRC) Framework regarding complex interventions with inspiration from O'Cathain et al [33,34]. The development process is reported following the GUIDance for the rEporting of intervention Development (GUIDED) recommendations [35].

With a user-centred approach and utilizing a co-creation process [36,37], stakeholders were recruited according to public involvement guidelines formulated by the International Collaboration for Participatory Health Research [38,39]. Such, stakeholders whose life or work is the subject of the research were onboarded in all phases of the research process to increase likelihood of acceptance by all represented parties involved. Stakeholder consent was informed, verbal, and documented by research-professionals. For stakeholders who were minors consent from parents/guardians were obtained. Consent was assessed by the Central Denmark Region's Committees on Biomedical Research (1-10-72-274-21).

## Setting

In Denmark, a gender-neutral and GP-based HPV-vaccination programme was implemented as part of the tax-financed Childhood Vaccination Program in 2019, after having included only girls since 2009. The vaccination is on parents' initiative after a governmental invitation when the child turns 12 years old.

The intervention was part of interventional research originated and developed in Aarhus. Aarhus is Denmark's second largest municipality with a population of, approximately, 360,000 of which 16% has ethnic minority background, dominated by non-western origin, especially Northern Africa and Middle East [40]. Aarhus was the base for stakeholder recruitment and the targeted intervention municipality.

## Implementation and analysis

The intervention will be conducted as a non-randomized implementation study and evaluated using both quantitative and qualitative analysis. Results are to be published in scientific articles. Quantitative outcomes will be HPV-vaccination coverage in the intervention-group compared to a four-fold larger control group. The control group will be comparable in age and community and will thus resemble the composition of ethnicity. Qualitative focus-groups are planned to include every sub-group affected by the intervention. Thus, adolescents, parents, school-professionals (teachers and leaders), and interveners are planned to be interviewed. By using quantitative as well as qualitative methods in the analysis, we can gain greater insight about intervention efficacy regarding acceptability, feasibility, and potential alterations for large scale implementation. The first round of implementation will be considered a pilot study [33].

## Target population

The target population is defined as the population that will potentially benefit from the intervention. That is, pupils and parents of 4th to 6th grade (ages 10–13) at municipally managed schools in Aarhus with a high proportion of pupils with an ethnic minority background, defined as >25% of the pupil proportion.

## Theoretical framework

The intervention development was framed by perspectives from ecological models of health behaviour [41,42], stating that several determinants affect health behaviour decisions, such as having your child vaccinated against HPV. These factors interact and, thus, interventions should be multi-level to encompass every layer of potential impact of decision making [43]. This approach was established from the pioneering stage of the development process throughout stakeholder recruitment and design, illustrated in Fig 1.

We leaned upon theories within public health and behavioural change. Most importantly the concept of health literacy, defined as the degree to which individuals have the ability to find, understand, and use information and services to make health-related decisions and actions [44,45]. Self-efficacy is defined as an individual's confidence in their ability to overcome perceived barriers [46]. Self-efficacy was included to understand the unfamiliarity, uncertainty and fear regarding HPV-vaccination for one's child perceived by parents, as well as potential nervousness for getting vaccinated perceived by adolescents. Further, The Health Belief Model state that an individual's course of action depends on perceptions of benefits and barriers related to health behaviour and this constituted the theoretical foundation [47]. In addition, social cognitive theory states that individuals learn from one another through watching behaviour and attitudes, and the results of those behaviours, empathizing the importance of a multi-level approach enhancing individual and local level of impact [48]. The joined theoretical framework guided the development process to achieve a high acceptability of developed intervention.

## Description of development process

The development process was iterative, but for the sake of understanding, illustrated as a forward moving process (Fig 2). It can be summarized in the following four phases exploited further in the following sections: 1) "Evidence synthesis of barriers and possible interventions" with exploration of barriers regarding HPV-vaccination within ethnic minorities. 2) "Acceptability of intervention" addressed through cooperation with stakeholders representing each level of influence for the intervention. 3) "Intervention elements and context" with the creation of a logic model with hypothecations of potential mechanisms of change. 4) "Co-production of intervention material" [49]. The development process took place from June 2020 to June 2022.

**Phase 1: Evidence synthesis of barriers and possible interventions.** Initially, a qualitative study on the perceptions of cervical cancer prevention among a group of ethnic minority women in Denmark [16] with three semi-structured focus-group interviews with ethnic minority women were conducted. These interviews explored barriers towards HPV-vaccination including user perceptions of feasible future interventions. From here, a review of literature was performed,

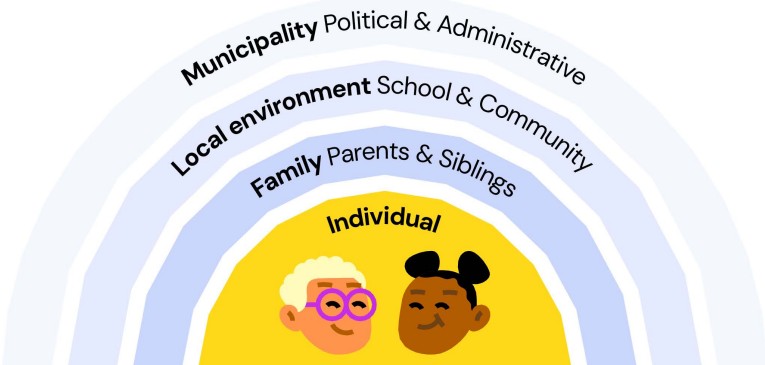

**Fig 1. Levels of intervention interest and impact.** Levels of potential impact of developed intervention with the purpose of changing health behavior and coverage of HPV-vaccination among children with an ethnic minority background.

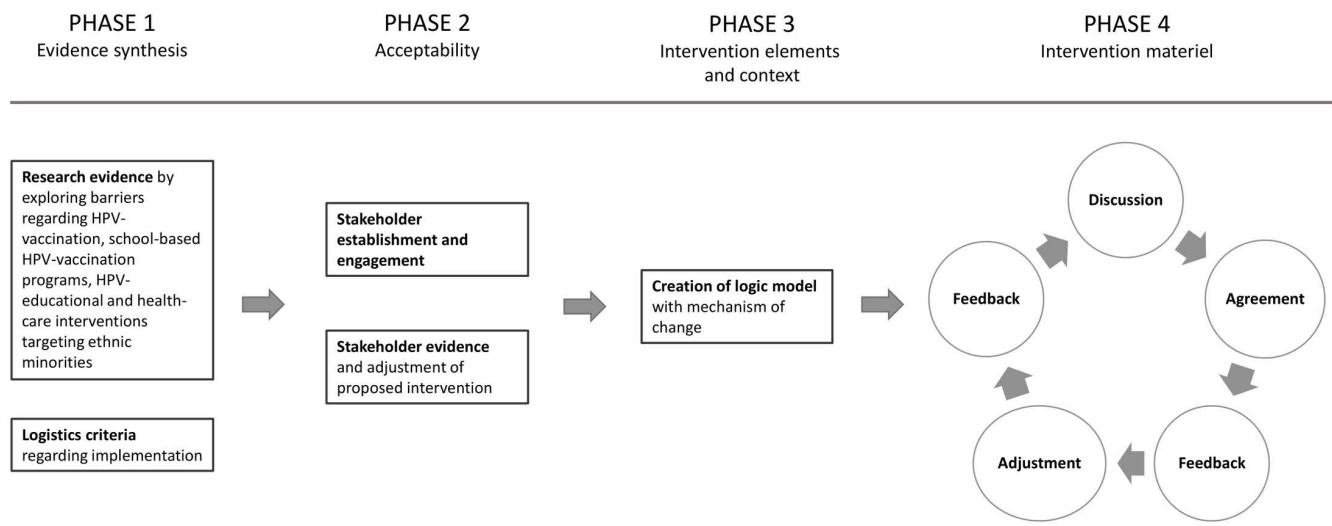

**Fig 2. Phases of intervention development.**

exploring school-based vaccination, HPV-educational programmes, and health-care interventions targeting ethnic minorities. This generated the evidence base.

**Phase 2: Acceptability of intervention.** Stakeholder recruitment was guided by our theoretical framework, ensuring representation from every level of interest and impact (see Fig 1). To impart interactions in the co-creation process between research-professionals and stakeholders, agreements regarding decision making were established from the outset.

The research professionals constituted the top decision-degree as they initiated the study [39,50]. Political and administrative stakeholders from the municipality were recruited with partnership-degree, as organizational and socio-political factors could have great influence on the development, and later the delivery of, the intervention. Close collaboration was established with a non-governmental organization, The Neighbourhood Mothers, mainly consisting of ethnic minority women of Northern African and Middle Eastern origin, representing the local community. Five of these women were recruited with consultation-degree and, around, 30 women from the same group with a feedback-degree. Two school-nurses were recruited with consultation-degree, representing the primary healthcare and local school environments that have a high proportion of pupils with an ethnic minority background. With feedback-degree representing the local and cultural environment, a gynaecologist with expertise in communicating gynaecological issues to ethnic minority women was recruited. Ten mothers and children aged 10–13 years with various ethnic and social backgrounds, were involved with a feedback-degree, representing the family and inter-personal environment.

Based on phase 1, the research team presented an initial proposal of the intervention to stakeholders with or above a consultation-degree for adjustment and approval before moving along to phase 3. Adjustments were made according to the primary principles for implementation as described in "theoretical framework".

**Phase 3: Intervention elements and context.** A logic model was made as a systematic and visual representation of intervention activities, hypothecations of potential mechanisms of change, and expected generated outcomes [51,52]. Inputs were the results from phase 1 and 2. Theory regarding potential unintended outcomes [53] and a description of resources needed for implementation were included in a work-edition of the logic model, not illustrated in this paper.

Context of the intervention may directly or indirectly affect the implementation and outcome of intervention. We included cultural, social, and local contexts as well as external factors.

**Phase 4: Co-production of intervention material.** This phase was characterized by a co-production approach [49] with a focus on feasibility and implementation. Stakeholders were involved in the development from form to content.

Material targeting adolescents was consulted with an organization specialized in communicating sexual-related issues to children, the Danish Family Planning Association [54], and an infographic specialist [55] regarding production. Material targeting parents was developed with a graphic specialist [56] and then translated to Somali, Turkish, Arabic and English by a professional bureau [57]. Native speaking stakeholders gave feedback and provided adjustments for the final translation.

## Results

In the following sections, results and contributions from the development process are further expanded upon.

### Results from phase 1: Evidence synthesis of barriers and possible interventions

The main barriers towards HPV-vaccination among ethnic minorities were unfamiliarity, uncertainty, and misconceptions of safety and relevance, as well as a fear of promiscuity if children were vaccinated against HPV. Also, uncertainty to GP's and the healthcare system as such were of influence. Practical barriers, such as language, booking systems and inconvenience were of likewise significance.

Interviewees stated that interventions should be tailored to ethnic minorities and focused on improving knowledge about HPV and HPV-vaccination for parents, as well as adolescents, in an effort to increase acceptability. A school-based intervention was proposed by interviewees and healthcare professionals as schools and school nurses constitute a trustworthy and reliable environment. We examined logistics criteria crucial for intervention implementation, finding that the timing of the intervention in relation to general school year planning was considered essential. An appropriate amount of time should be reserved for school and local preparation as well as competence development for interveners (intervention providers), with a strict project management and knowledge according to the pathways of communication [58].

Most importantly, the intervention should be intuitive and require the fewest possible resources from school-professionals, interveners, and overall, with a time constraint. Further, guidelines for interveners was needed to ensure clear communication [59].

Evidence showed that intervention materials and communication needed to be uniform, simple, in ethnic minority languages, and inform about the safety and efficacy of HPV-vaccination [31]. If possible, the information would be best adopted if presented visually, with an opportunity for repetition [60,61]. Further, nervousness is an essential consideration for adolescents regarding vaccination [62] and it is important to address this and to include adolescents in healthcare decisions [28,63].

### Results from phase 2: Acceptability of intervention

An intervention proposal to stakeholders was comprised of three equally weighted elements in a school-setting: HPV-education targeting parents, HPV-education targeting adolescents and HPV-vaccination during school hours. The intention was to promote better understanding, reduce fear and misconception, and to make HPV-vaccination accessible with minimum impact on school resources.

The proposed intervention was agreed upon with minor revisions from stakeholders ensuring minimum resource consumption from schools and their personnel, and included an agreement to change the wording from education to information in the material and communication. It was highlighted by school-professionals, that HPV-vaccination at school should be clearly stated as an offer that was proposed by healthcare professionals and not by the schools and their personnel. School-nurses emphasized the importance of ownership for interveners, as well as sustainability in the intervention's material.

With an insight knowing that ethnic minority parents participate less in meetings/access less information given by the schools, it was a vision from the stakeholders to do personal outreach by local ambassadors to gain optimal contact with parents. However, as one of the primary principles for the intervention was to focus on implementation in an everyday context with minimal resource consumption, personal outreach was ultimately excluded.

## Results from phase 3: Intervention elements and context

The developed logic model is shown in Fig 3. The possible mechanism of change was formulated as accommodation to barriers regarding HPV-vaccination as well as enhancing health literacy and self-efficacy. Assumptions of contextual factors were a statement of an anti-first mover environment, and hence a need for local ambassadors and peer-support. In the cultural context there was an assumption of high trust to volunteers and school professionals and, thus, a broad presentation of ethnicities and authorities working together was hypothesized to increase acceptability. External factors, like the COVID-19-pandemic, guided the development in creating materials that were suitable for implementation during potential lockdowns.

As a result of the evidence base and stakeholder contribution, we hypothesized that both parental and adolescents' attitudes towards HPV-vaccination could be altered and acceptability could be increased, with more awareness of HPV-diseases, perception of relevance of prevention, and further positioning the offer of HPV-vaccination at school as being convenient and accessible. For adolescents, we hypothesized that increased self-care would be of significance, as well as interaction and peer-support between adolescents at school [26,64–66]. Concerning the cultural context, we hypothesized that storytelling about preventing disease would be more effective than focus on HPV being sexual transmitted.

**Outcome and evaluation strategy.** We hypothesized that an intermediate effect would be intervener-parents acting as ambassadors in local environments. As a result of this, the intervention could potentially lead to less taboos related to HPV and HPV-vaccination, improved perception of relevance of HPV-vaccination, and overall, increase accessibility of information and vaccination. In the long term, we hypothesized an increased coverage of HPV-vaccination in the target group and later, the local environment. If this is the case, we would expect to see an enhanced political request for tailored HPV-solutions to reduce inequity.

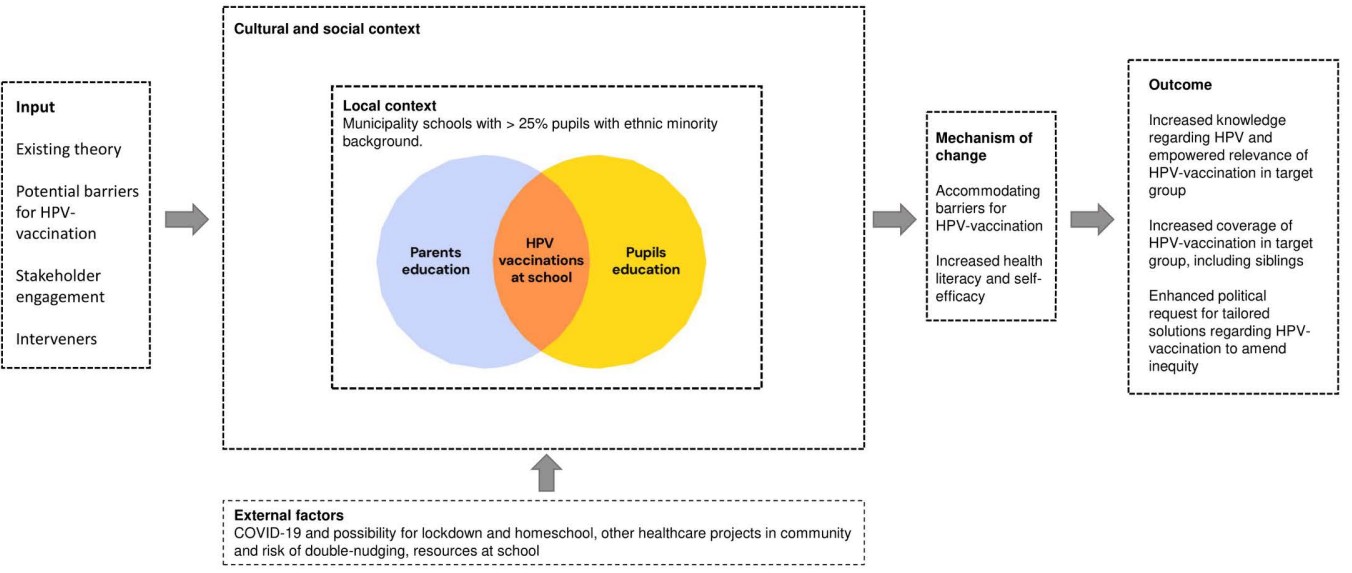

**Fig 3. Context of intervention Logic Model.**

Potential unintended outcomes were identified as clustering anti-vaccination behaviour and fear of change in sexual behaviour in adolescents, as perceived by parents. To prevent this, we aimed to be reliable in storytelling and messaging.

Evaluation was of importance to stakeholders. They were interested in, not only a quantitative result concerning HPV-vaccination coverage, but also a thorough analysis of how the intervention was perceived by target group as well as all professionals affected by intervention. A mixed-method analysis was chosen and is described in the section titled "Implementation and analysis".

### Results from phase 4: Co-production of intervention material

Generic material was developed, thus less dependent on individual variation of resources at the intervention sites. Material was designed to neutrally explain the benefits adolescents would receive from HPV-vaccination, equally for girls and boys. Stakeholders emphasized, especially those with an ethnic minority background, the importance of a personal, relatable message rather than statistics heavy information and one stated the importance of a clear message: "relevant for me and my child".

Material targeting parents was a manuscript that describes vaccine safety and efficacy if given to adolescents before sexual debut, explaining HPV virus transmission. The manuscript was designed to be given orally to parents at school-parental meetings and subsequently video to be revisited by parents if needed. The one providing this messaging is a Medical Doctor, in an effort to increase trust.

Material targeting adolescents was mainly a developed animation, which consist of a conversation between a boy and a girl, each with a different ethnic background, talking and asking each other questions about HPV and HPV-vaccination. It was a vision to illustrate communication coming from adolescents/equals to viewers in a known language with familiar terms. The overall message is that adolescents can be vaccinated against HPV and protected against HPV-related cancer as adults. It does not contain information about HPV being a sexually transmitted virus but does inform that it is a virus transmitted between adults and can, potentially, result in diseases, such as cancer, when they grow-up. It addresses nervousness related to vaccination and potential acute side-effects. Various ethnicities and religions are included to accommodate the importance of target adolescents feeling represented. At the end, the two animation-adolescents address the viewer, stating that now the viewer knows something about HPV and HPV-vaccination, and that, together with parents, they can decide if they should get vaccinated. This to promote a discussion and empower the adolescents as well as promote the importance of a knowledge pathway from adolescent to parent. To accommodate potential questions asked by adolescents regarding transmission, we included preparation materials for interveners related to such questions, emphasizing that this information should not be withheld, it just should not be the focus.

Overall, the developed material differs from material that is already available on the subject by including simpler language and utilizing some minority languages. The material is predominately video and animation instead of written text. There is no mention of sexual transmission when targeting adolescents and there is a general focus on prevention of disease by vaccination. Further, it includes an italicization of nervousness and a presentation of peer-support, as well as a request to have a conversation about vaccination between adolescents and parents to support the known pathway of communication. In addition, adolescents from different minorities are visually represented.

### Final intervention: "Two shots for life. HPV-vaccination at your school"

The template for the Intervention Description and Replication (TIDieR) guidelines is used as a thorough description of the developed intervention and its elements (Table 1). The core intervention elements and an overview of the material are illustrated by Fig 4. Three core elements are parents' education beneficial for parents, pupils' education beneficial for adolescents, and HPV-vaccination at school as beneficial for both groups.

**Table 1. Details of intervention "Two shots for life. HPV-vaccination at your school", according to TIDieR template.**

| Item | Description |
|---|---|
| 1. Brief name | "Two shots for life. HPV-vaccination at your school".<br>Co-creation of a complex intervention offering HPV-education and -vaccination at public schools. |
| 2. Why | Ethnic minority children have a lower coverage of HPV-vaccination due to practical, cultural and emotional barriers. This intervention aims to overcome barriers for HPV-vaccination by combining school-based HPV-education and –vaccination in a targeted intervention. |
| 3. What materials | **Parents education**<br>- A one-page HPV and HPV-vaccination-information and offer of HPV-vaccination at school available in Turkish, Arabic, Somali, English or Danish.<br>- Submission to vaccination online at a REDCap survey (available in Danish or English). Submission/consent form also possible via a written one-page available Turkish, Arabic, Somali, English or Danish.<br>- An oral presentation (15 min) explaining HPV and HPV-vaccination and practical information about offer of HPV-vaccination at school.<br>- A simple roll-up assisting oral presentation.<br>- A video texted in Turkish, Arabic, Somali, and English with an oral presentation explaining HPV and HPV-vaccination, presented by medical doctor.<br>- Facebook group solely for parents included in the intervention. Possibility for contact with project manager and availability of video, animation and one-page information.<br>**Pupils education**<br>- An animation (3 min), illustrating facts about HPV and HPV-vaccination as well as addresses nervousness. Available at Facebook group.<br>- A tote bag with intervention name and logo including postcard.<br>- A postcard supporting facts given in the animation and with an additional opportunity to submit for participation by QR-code linking to REDCap survey.<br>- A poster for classrooms with facts about vaccination-date, referral to project group regarding questions and three good advice concerning vaccination day (remember breakfast, if nervous talk to a friend or an adult, if questions talk to an adult)<br>- sticker with intervention name and logo for distribution at school.<br>**Interveners (medical doctor, school nurses and vaccinators)**<br>- A guideline with supporting information about HPV, HPV-vaccination and project information.<br>- 5-question cards available for school-nurses as conversation-support.<br>- A guideline for vaccinators regarding vaccination-procedure and medical conditions and treatment in relation to vaccination (vasovagal reaction, anaphylaxis and stabbing injury).<br>**Additional personal**<br>- An overview of intervention elements and timeline of intervention during schoolyear for schoolteachers and school administrator with referral to project group if questions.<br>- Project Information for awareness and referral to project group if questions to general practitioners in the municipality.<br>Contact to authors for further information and availability of material, including video and animation. |
| 4. What procedures | **Preparation**<br>- Establishment of collaboration between municipality, school leaders and interveners.<br>- Preparation of interveners.<br>- Information about HPV-vaccination at school given to school-professionals through school-communication platform before intervention start.<br>**Parents education**<br>- Information about offer of HPV-vaccination at school next school year at the end of preceding school year via school-communication platform.<br>- One-page information and submission to all parents in 4th-6th grade via digital, personal mailbox at intervention start.<br>- Information of attendance by medical doctor at school-parental meeting through school-communication platform.<br>- At first school-parental meeting a short oral presentation (15 min) explaining HPV and HPV-vaccination is given by medical doctor accepting questions/remarks by parents.<br>- One-page information and submission/consent form available at school-parental meeting.<br>- Local ambassadors attend school-parental meeting.<br>- All parents not attending school-parental meeting receives one-page information and submission/consent form for participation by letter with a stamped return-envelope.<br>**Pupils education**<br>- In a school-nurse setting the adolescents watch animation followed by a talk, supported by questions-card if needed.<br>- The adolescents receive a tote bag with postcard.<br>- Posters are displayed in classroom.<br>**HPV-vaccination at school**<br>- HPV-vaccination dates are organized with school personal and communicated to parents through school communication platform.<br>- Before vaccination dates submission forms are registered in REDCap and checked for correct information and validity.<br>- Five months between first and second dose of vaccination, making first dose eligible for fall and second dose eligible for spring during a schoolyear.<br>- Vaccination interveners (healthcare professionals), always with one medical doctor present for diagnosis and treatment of potential anaphylaxis.<br>- Vaccinations registered in The National Danish Childhood Vaccination Register, or appropriate registration according to existing guidelines. |

*(Continued)*

**Table 1.** (Continued)

| Item | Description |
|------|-------------|
| 5. Who provided | - **Preparation** is generated by project manager(s).<br>- **Parents education** delivered by medical doctor and volunteering local ambassadors (women with ethnic minority background).<br>- **Pupils education** delivered by local school nurse, prepared by written information and one online preparation meeting by project manager(s).<br>- **HPV-vaccination** delivered by medical professionals with one logistic manager and two vaccinators, one of the three being a medical doctor.<br>- Communication through school communication platform generated by project manager(s) and local school administrator. |
| 6. How | **Preparation**<br>- Meetings, in person or online as preferred by collaborators, interveners implementation sites.<br>**Parents education**<br>- School-parental meetings divided in years at local school, e.g., all parents of 4th graders at one school.<br>**Pupils education**<br>- Class wise, boys and girls together. The school nurses are allowed to separate into groups of sex if assessed necessary.<br>- Animation showed on big-screen.<br>**HPV-vaccination at school**<br>- Typical school day, respecting breaks.<br>- Vaccinators and logistic manager visit all classes in the morning to greet the adolescents and to ensure logistic-agreement with schoolteacher.<br>- Class by class adolescents are followed by logistic manager to vaccination-site at school.<br>- Two-and-two adolescents are vaccinated behind dividing walls, with the rest of the group waiting turns or resting after vaccination (15 min).<br>- Juice, fruit, biscuits etc. available after vaccination.<br>- Logistic manager ensures vaccination registration. |
| 7. Where | Intervention developed for municipality schools with >25% of pupils with an ethnic minority background, originally in Aarhus, Central Region of Denmark. |
| 8. When and how much | Intervention set for implementation across schoolyear from August until June.<br>**Parents education**<br>- School-parental meeting already planned by school, typically in August/September allowing 15 minutes to parents' education.<br>**Pupils education**<br>- Pupils' education in September/October after parents' education. School-nurses plans session with schoolteachers scheduling with a duration of 30 minutes. In 5th grade the pupils' education is in extension of already scheduled puberty education.<br>**HPV-vaccination at school**<br>- First dose of HPV-vaccination in November/December, planned with one day pr. year (e.g., one day to all 4th graders at one school)<br>- Followed by second dose of HPV-vaccination in May-June, planned with one day pr. year.<br>- 1 follow-up day pr. vaccination-round. |
| 9. Tailoring | Intervention design, core-elements and material are tailored to described target-group, ensured by a high stakeholder involvement and co-creation approach.<br>Individual tailoring to schools, school nurses and classes are accommodated by project manager(s) in the extend that core-elements are delivered and safety to vaccination maintained. |
| 10. Modifications | Not applicable.<br>Scheduled time for second dose of vaccination can be adjusted, concerning time-evaluation after first dose of vaccination. |
| 11. How well – planned | A thorough cooperation between school professionals and interveners should ensure each core element are delivered in conformity with intervention design and feasible at each intervention school.<br>A clear communication and visibility of project manager(s) regarding eventual questions by interveners or school personal, as well as parents. |
| 12. How well – actual | Not applicable. |

The final communication strategy has a primary focus on health prevention in terms of vaccination, rather than disease, virus, and sexual transmission. The graphics were designed to be innocent, bright, and simple, with the aim of being appealing to adolescents. Decision making is assigned to parents but aims to increase health literacy of adolescents while considering the effect of peer-support and group conformity.

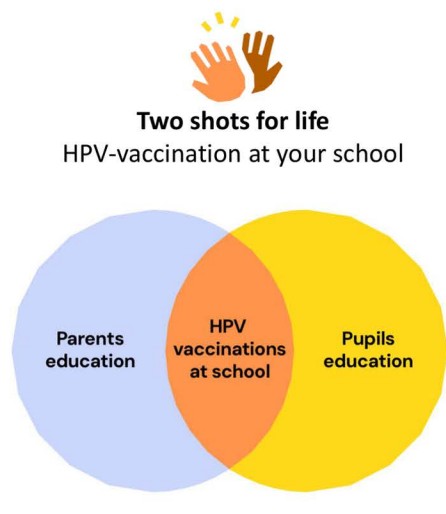

**Parents**

**School-parental meeting: 15 minutes oral presentation of HPV, HPV-vaccination and offer of HPV-vaccination at school by healthcare professional**

▶ Written information available in Turkish, Arabic, Somali, Danish and English

▶ Registration and consent form available in Turkish, Arabic, Somali, Danish and English

**Social Media Platform**

▶ Video of healthcare professional giving oral presentation of HPV and HPV-vaccination. Texted in Turkish, Arabic, Somali, Danish and English

**Written reach out for non-attenders to school-parent meeting**

**Online registration and consent form**

**Pupils**

**Poster in classroom**

**School-nurse setting**

▶ Animation (3 minutes). A conversation between a girl and a boy talking about HPV and that it is a good idea to get vaccinated as a child. Addresses nervousness for vaccination and represents various ethnicities.

▶ Classroom talk about HPV and HPV-vaccination

▶Take-home tote bag including a postcard with QR-code for online registration and consent form

**Other materials**

**Guideline for intervention providers**

**Information poster for teacher's room**

**Information for general practitioners**

**Fig 4. "Two shots for life" logo, core elements and overview of material.**

Material example of animation is attached in supporting information S1 File.

## Preparation of interveners and local environment

Intervention management is allocated to a few project-managers to ensure strict management and easy access for interveners and local environment, a high level of trust, and uniform implementation of the logistical requirements. Intervener school-nurses is prepared by one group-meeting and a one-to-one meeting with project manager. They are supplied with research-based information about HPV and HPV-vaccination, advice, and support to facilitate the intervention with creation of a safe space through a guideline for intervention implementation. Intervener vaccinators are nurses or medical doctors, recruited at local Hospital and prepared by written information and guideline for vaccination.

School leaders are prepared at a group-meeting and a one-to-one meeting, and subsequently act as suppliers of main information to schoolteachers. The GPs in the municipality, are contacted with written information for awareness of ongoing intervention. Local ambassadors are informed at local network meetings [Neighbourhood Mothers].

## Ethical considerations to intervention, vaccination consent and security

The intervention protocol was initially assigned at clinical trials.gov [67]. The intervention was designed as a behavioural intervention, as HPV-vaccination with Gardasil9 [68] is transferred from the GPs setting to local schools. The intervention will be conducted in accordance with the Good Clinical Practice Guidelines and reported to the Data Protection Agency in Central Denmark Region (ref.nr. 761330, 1-16-02-494-21) and Central Denmark Region's Committees on Biomedical Research (1-10-72-274-21).

Registration and consent to vaccination will be by holder of parental authority. This will be double-checked with given consent information and personal data of intervention group supplied by the Municipality. This ensures adolescents are not able to sign themselves up for vaccination. If any mismatch in given information, parents will be contacted to confirm consent. Vaccination will be conducted according to medical guidelines under medical doctor surveillance with possibility of diagnosis and treatment of anaphylaxis. Vaccination will be registered in The National Danish Childhood Vaccination Register. All data are handled according to the EU's General Data Protection Regulation. Cooperation between research team and municipality have been registered as an official cooperation agreement.

## Discussion

This paper describes the development of a co-created complex intervention aiming to increase HPV-vaccination coverage among adolescents with an ethnic minority background. The final version of "Two shots for life" consists of school-based HPV-education and -vaccination targeting both adolescents and their parents at public schools with a high proportion of pupils with ethnic minority background.

### Communication focus

Focusing on prevention of potential disease by vaccination rather than the aetiology of HPV-related disease by sexual transmission, might decrease the fear of HPV-vaccination leading to promiscuity, which represents a barrier not only among ethnic minorities [31], but across ethnicities [69,70]. Focusing on the positive outcome and efficacy, rather than potential risk could be effective since safety regarding HPV-vaccination is an important message in HPV-vaccination communication [71–73]. In a study aiming to develop a cultural-sensitive information-video to facilitate cervical cancer screening, means to protect one's child's health could be a potentially effective narrative perceived by the target group with a religious background in Islam [60].

One of the studies found that information about sexual transmission of HPV to adolescents aged 13–15 years did not alter their HPV-vaccination acceptance [69]. Our intervention group is somewhat younger and potentially more immature regarding sexual awareness and knowledge. Also, the influence and importance of ethnic and cultural background is not part of the referred study.

Communication strategy was developed to overcome barriers for ethnic minority citizens, but many of these barriers are also present for parents with ethnic majority background, potentially giving the unintended, but positive outcome that children of various ethnic or cultural background will benefit from the intervention.

### Communication providers

When communicating health information, the cross-cultural context often challenges the encounter by differences in language and an authoritarian approach, blurring messages of HPV-vaccination relevance. A Cochrane study have shown that face-to-face interventions may be more effective than written information regarding educating parents about vaccination to achieve an increased knowledge and intention to vaccinate [74]. Education elements in developed intervention are face-to-face delivered, exploiting a provision of trust, emphasized as being an important facilitator in studies exploring ethnic minority women's perceptions of potential interventions [16,17]. By making information personal and accessible in multiple languages and format known to increase accessibility, we have tried to emphasize elements supported by studies exploring strategies for cross-cultural communication [30,75].

Studies have shown that communication about HPV-vaccination should be given by health care professionals with an ability to address potential emotional barriers and hesitation [59,76]. In a Swedish, qualitative study exploring healthcare providers' (school nurses) perception of barriers in a national school-based HPV-vaccination programme, it was enhanced that training and guidance of vaccination professionals is crucial to accommodate vaccine hesitancy

and barriers for HPV-vaccination [77]. In "Two shots for life", interveners will be prepared with information about the intervention overall being targeted to accommodate barriers among ethnic minority families, but individual preparation to accommodate barriers as intervention providers is not part of the final design of intervention. Educational core elements were developed to communicate generic information with an already set narrative in group settings (parental meetings and class-setting). The overall generic education approach could be a weakness potentially leading to individual barriers not being accommodated but was chosen to ensure an equal and sustainable implementation strategy across sites.

### School-based setting

Studies have tested effectiveness of school-based HPV-vaccination interventions [24,66], and few have developed and some tested HPV-education interventions and their influence on HPV-vaccination coverage [24,78–80]. One of these is ongoing and targets schools with low uptake in a national school-based HPV-vaccination programme, adding a co-produced educational package [79]. A multi-phased approach like ours to overcome barriers among ethnic minorities [31,80] by combining described elements in an intervention has not yet been completed.

By a school-based approach, "Two shots for life" leans upon the social support theory that emphasises a vaccination-setting consisting of social support, compassion, and reassurance can provide self-worth, development of sense of control and reduction of uncertainty. Potentially, school setting gives a great effort at individual level with a minimal cost at society level. How far into social sectors health interventions can be allowed to exploit, should be assessed, and a cost-benefit analyses of our developed intervention will be assessed in the planned quantitative analysis.

By intervening in a school setting as health authorities, the encounter is trustful, and school-nurses as interveners provides thorough knowledge regarding best possible communication setting [81]. A result from phase 3 was the change of wording from 'education' to 'information' throughout implementation and intervention communication. This, to respect teacher's profession and position as suppliers for education in school setting. Such small, yet important alteration made to accommodate implementation sites were shown to be of importance when crossing sectors.

### Targeting both adolescents and parents

A need for educational core elements addressing adolescents as well as parents was surfaced in phase 1–2, with a combined education effort potentially providing a synergy when addressing health literacy. Studies have shown positive effects of HPV-education of adolescent in a school-based HPV-vaccination programme [24,26,78], in the same scene investigating parent's knowledge and attitude towards the HVP-vaccination programme, as well as the education intervention, finding perceptions to be overall positive and education of both groups relevant [23]. An Australian, schools- based HPV-education intervention targeting adolescents, found that the intervention was not effective in increasing HPV-vaccination uptake, possibly due to an already high vaccination-coverage. But it did improve knowledge and psychosocial outcomes [78] enhancing the long-term potential by positively altering health literacy and self-efficacy leading to a change in behaviour and hence, HPV-vaccination uptake.

Most available literature regarding vaccination-coverage and barriers does not cover both sexes, but is dominated by research focusing on girls, since HPV-vaccination started out only being available for the female sex. Now, HPV-vaccination being part of the gender-neutral national vaccination programme, we also made our developed intervention gender-neutral. One could argue the evidence basis being somewhat weak concerning unknown barriers for boys-parents and among male adolescents themselves.

### Strengths and limitations

The development process described has both strengths and limitations. The approach with co-creation and an overall alignment with complex intervention framework proposed by MRC ensured context analysis and consideration,

identification of key uncertainties, and investigation of existing knowledge and theories in the research field. Involvement of stakeholders with representation of target group as well as intervention deliverers in a co-creation process and co-production of intervention materials has been shown to ensure ownership in research, as well as improvement of adaption and tailoring to the specific context [37]. As the complex intervention framework emphasizes, the process has been iterative with possibility to go back and revisit preceding phases of the development process.

Involvement of stakeholders from every layer of potential impact on decision making regarding HPV-vaccination in a school- and a culturally diverse context have resulted in insights and accommodation to otherwise unknown barriers for feasibility and implementation that a solely academic-led approach could not have provided. The collaboration across sectors can potentially positively impact outcome for the local delivery of the intervention. A limitation is onboarding of stakeholders carried out by healthcare professionals and researchers, potentially overlooking key-stakeholders. In a Swedish, qualitative study, school nurses expressed the importance of having close cooperation with teachers in order to be successful in their health-promoting task [81], enhancing a potential weakness by not having involved teachers as stakeholders in our developed intervention.

The time-consuming condition of the development process when following the MRC complex intervention framework and an overall co-creation process, is broadly acknowledged. However, it should be enhanced for future project planning if aiming to alter health behaviour in a multi-level approach and complex setting.

Transparency of our development process allow others to replicate and gain insights into the learning process of developing "Two shots for life" and additionally, detailing the development process should add to the understanding of later evaluation of outcomes.

Next step is to test the intervention regarding effectiveness on HPV-vaccination coverage, its feasibility, and acceptability by target group, shedding light on the potential of bringing health-interventions into school systems.

## Conclusion

The development process of a co-created intervention aiming to increase HPV-vaccination coverage among adolescents with ethnic minority background led to a school-based HPV-education and -vaccination intervention tailored to accommodate known barriers towards HPV-vaccination. Involvement of stakeholders of this multi-level intervention has covered an individual, family, cultural, structural, and political context. Communication focus was emphasized to be on HPV-vaccination as a prevention effort rather than HPV being sexual transmitted. Communication strategy was developed to overcome barriers among ethnic minority citizens, but potentially accommodates barriers for HPV-vaccination across ethnicities. Presenting a thorough development process with transparency, we hope to enhance future development of other similar complex interventions.

## Supporting information

**S1 File. Material example Pupils education animation in still pictures.**
(TIF)

## Acknowledgments

We are most thankful to all collaborative partners in the Municipality of Aarhus and the Neighbourhood Mothers as well as stakeholders in the project for their indispensable contribution and valuable input.

## Author contributions

**Conceptualization:** Anne Katrine Leonhard, Sara Koed Badre-Esfahani, Mette Larsen, Lone Kjeld Petersen, Lene Seibæk.

**Data curation:** Anne Katrine Leonhard, Lene Seibæk.

**Formal analysis:** Anne Katrine Leonhard, Sara Koed Badre-Esfahani, Lene Seibæk.

**Funding acquisition:** Anne Katrine Leonhard, Sara Koed Badre-Esfahani, Mette Larsen, Lone Kjeld Petersen, Lene Seibæk.

**Investigation:** Anne Katrine Leonhard, Sara Koed Badre-Esfahani.

**Methodology:** Anne Katrine Leonhard, Sara Koed Badre-Esfahani, Mette Larsen, Lene Seibæk.

**Project administration:** Anne Katrine Leonhard, Lene Seibæk.

**Resources:** Anne Katrine Leonhard.

**Supervision:** Mette Larsen, Lone Kjeld Petersen, Lene Seibæk.

**Validation:** Anne Katrine Leonhard, Lene Seibæk.

**Visualization:** Anne Katrine Leonhard, Mette Larsen.

**Writing – original draft:** Anne Katrine Leonhard, Lene Seibæk.

**Writing – review & editing:** Sara Koed Badre-Esfahani, Mette Larsen, Lone Kjeld Petersen, Lene Seibæk.

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
