## [Decision Letter · Decision Letter 0]

20 Aug 2024

PONE-D-24-14821“Two shots for life. HPV-vaccination at your school”. Co-creation of a complex intervention to reduce ethnic inequity in childhood HPV-vaccination in Denmark.PLOS ONE

Dear Dr. Larsen,

Thank you for submitting your manuscript to PLOS ONE. After careful consideration, we feel that it has merit but does not fully meet PLOS ONE’s publication criteria as it currently stands. Therefore, we invite you to submit a revised version of the manuscript that addresses the points raised during the review process.

We look forward to receiving your revised manuscript.

Kind regards,

Maria Lina Tornesello

Academic Editor

PLOS ONE

Journal Requirements:

3. Please include your tables as part of your main manuscript and remove the individual files. Please note that supplementary tables (should remain/ be uploaded) as separate ""supporting information"" files

Reviewers' comments:

Reviewer's Responses to Questions

**Comments to the Author**

1. Is the manuscript technically sound, and do the data support the conclusions?

Reviewer #1: Partly

2. Has the statistical analysis been performed appropriately and rigorously? 

Reviewer #1: N/A

3. Have the authors made all data underlying the findings in their manuscript fully available?

Reviewer #1: Yes

4. Is the manuscript presented in an intelligible fashion and written in standard English?

Reviewer #1: Yes

5. Review Comments to the Author

Reviewer #1: This is a well written manuscript describing the development of a school-based complex intervention to promote HPV vaccination to minority families in Aarhus, Denmark. The work takes into account the relevant guidelines and demonstrates this explicitly. One major weakness of the report is that (as found in other comparable projects), one has the impression that the evidence-based justification and development of the format (=> accessibility) is more important than the actual content (=> acceptability). I ignore whether this is a problem with the actual project or only the report – in any case, I suggest that the authors strengthen the content side.

L55: please state to which regions/countries this statement about inequalities applies to.

L69-72: Please revise the sentence to be more precise: which elements have not yet been addressed in studies and will be addressed here ?

L89 and following: Please describe in the setting or other part of the introduction/methods which was the context for this complex intervention: eg, was this interventional research or a routine public health intervention? How was it justified – ie, was there are specifically low coverage in Aarhus or Denmark, please show coverage estimates to support the need for targeted action. Was the intervention supposed to be rolled out over Denmark, following this development and evaluation? Also, please describe the profile of the minority population in Aarhus, demographically and among participants of the co-construction.

L181: The authors describe in the methods that the pre-existing literature concerned cervical cancer prevention in minority women in Denmark. This is only partially relevant for the field of gender-neutral HPV vaccine promotion for minority adolescents. I would therefor recommend developing in more detail the results of the evidence basis, as they were obtained through the specific actions of this project. They should be presented in more structured way and separating elements on adolescents, mothers, fathers … ; on accessibility and acceptability.

As acceptability among minority families does not appear as a main strategic point in this work, please justify more clearly that acceptability was not a main barrier in the targeted population. If this is not the case, both methods and results would need to be revised to explain the underlying concepts (eg, 5C or 7C models are standard nowadays) and intervention levers (use of nudge, any consideration for frequent misconceptions) that were used to address low acceptability. For example: families often think that the child is too young to be vaccinated. If this is the case in the Aarhus minority population, as well, what specific strategy was used to avoid this perception being a barrier to vaccination?

Similarly, L204: the authors mention the need to reduce misconceptions, but this neither misconceptions nor barriers to vaccine acceptance are mentioned in phase 1. Please address these points more explicitly.

L201 and following: Acceptability of vaccination could be a main barrier

L231 and following: the authors write that they hypothised that not speaking about HPV as an STI would be more effective. This is a quite weak basis for a key element of the communication strategy, which even is mentioned in the conclusion (L446), in the absence of any evaluation. Please develop the justification for this hypothesis!

L271: Please summarise the key points in which the final material was substantially different from standard material that is used in Denmark for information on HPV vaccination to adolescents. I have the impression that this difference is simpler language and use of minority languages, no mention of STI, and representation of adolescents from different minorities. This impression may be simplistic, so please provide the summary.

L300-326 should be moved further up, as this information was missing in the beginning to understand the context and scope of the present report. Please describe the role of national, regional and local health and school authorities in commissioning or collaborating in this project.

L331: I’m wondering here whether school staff was involved (this is addressed later), and if not, why.

The discussion (L356-410) is lengthy and presents elements that I would have hoped to find in the description of the co-construction (evidence). I suggest reorganising the manuscript to bring to the description of the development the specific relevant elements on how which evidence was used.

Please have a revision of the English language of the manuscript.

6. PLOS authors have the option to publish the peer review history of their article (what does this mean?). If published, this will include your full peer review and any attached files.

Reviewer #1: No

---

## [Author Response · Author response to Decision Letter 1]

9 Jan 2025

Dear Academic Editor Maria Lina Tornesello,

Thank you for the opportunity to revise and resubmit our manuscript. We appreciate the comments and suggestions, and we have made changes to the manuscript accordingly.

Below you will find a point-by-point response to the reviewers' comments. First column is copies of the reviewers’ comments; our answers are in second column.

We hope that you find our changes satisfactory. Please do not hesitate to contact us in case of clarifying questions.

We confirm that our submission contains all raw data required to replicate the results of your study. The manuscript is not based on numeric data.

A marked-up copy of the manuscript with tracked changes has been uploaded as well as an unmarked version without tracked changes.

Kind regards,

Anne Katrine Leonhard

---

## [Decision Letter · Decision Letter 1]

11 Mar 2025

PONE-D-24-14821R1“Two shots for life. HPV-vaccination at your school”. Co-creation of a complex intervention to reduce ethnic inequity in childhood HPV-vaccination in Denmark.PLOS ONE

Dear Dr. Larsen,

Thank you for submitting your manuscript to PLOS ONE. After careful consideration, we feel that it has merit but does not fully meet PLOS ONE’s publication criteria as it currently stands. Therefore, we invite you to submit a revised version of the manuscript that addresses the points raised during the review process.

We look forward to receiving your revised manuscript.

Kind regards,

Maria Lina Tornesello

Academic Editor

PLOS ONE

Journal Requirements:

Reviewers' comments:

Reviewer's Responses to Questions

**Comments to the Author**

1. If the authors have adequately addressed your comments raised in a previous round of review and you feel that this manuscript is now acceptable for publication, you may indicate that here to bypass the “Comments to the Author” section, enter your conflict of interest statement in the “Confidential to Editor” section, and submit your "Accept" recommendation.

Reviewer #2: (No Response)

2. Is the manuscript technically sound, and do the data support the conclusions?

Reviewer #2: Yes

3. Has the statistical analysis been performed appropriately and rigorously? 

Reviewer #2: N/A

4. Have the authors made all data underlying the findings in their manuscript fully available?

Reviewer #2: Yes

5. Is the manuscript presented in an intelligible fashion and written in standard English?

Reviewer #2: Yes

6. Review Comments to the Author

Reviewer #2: The manuscript entitled "“Two shots for life. HPV vaccination at school”. Co-creation of a complex intervention to reduce ethnic inequality in childhood HPV vaccination in Denmark." by Leonhard AK et al., reports the overall strategy and specific steps to increase acceptance of HPV vaccination in ethnic school children through the involvement of all stakeholders, including parents, who must give written consent. The relevance of the study is the possibility of extending the described strategy to other international contexts.

Minor suggestions are:

1. Line 50 “the most preventive forms of cancer“ should be changed to “the most preventable forms of cancer”;

2. Lines 106 and 304: It would be appropriate to further elaborate the 4 -folf control group: specifying whether it will be of the same ethnicity and time period in schools not involved in the study, where vaccination adherence could be higher than in the past due to other dissemination programs, including social media.

7. PLOS authors have the option to publish the peer review history of their article (what does this mean?). If published, this will include your full peer review and any attached files.

Reviewer #2: **Yes: **Franco M Buonaguro, M.D.

---

## [Author Response · Author response to Decision Letter 2]

7 Apr 2025

Dear Academic Editor Maria Lina Tornesello,

Thank you for the opportunity to revise and resubmit our manuscript. We appreciate the comments, and we have made changes to the manuscript accordingly.

Below you will find a point-by-point response to the reviewers' comments. First column is copies of the reviewers’ comments; our answers are in second column.

We hope that you find our changes satisfactory. Please do not hesitate to contact us in case of clarifying questions.

We confirm that our submission contains all raw data required to replicate the results of your study. The manuscript is not based on numeric data.

A marked-up copy of the manuscript with tracked changes has been uploaded as well as an unmarked version without tracked changes.

Kind regards,

Anne Katrine Leonhard

Line 50 “the most preventive forms of cancer“ should be changed to “the most preventable forms of cancer” Thank you, we have changes line 50 according to proposed linguistic revision.

Lines 106 and 304: It would be appropriate to further elaborate the 4 -folf control group: specifying whether it will be of the same ethnicity and time period in schools not involved in the study, where vaccination adherence could be higher than in the past due to other dissemination programs, including social media. Thank you – agree.

We have elaborated the 4-fold control group in line 106. “The control group will be comparable in age and community, and will thus resemble the composition of ethnicity.”

Further discussion and clarification on control group, comparable considerations as well as statistical will be discussed in a future article revolving quantitative results of intervention.

We cannot see the control-group mentioned in other phrases of lines in the manuscript, or line 304.

---

## [Editor Report · Decision Letter 2]

16 Apr 2025

“Two shots for life. HPV-vaccination at your school”. Co-creation of a complex intervention to reduce ethnic inequity in childhood HPV-vaccination in Denmark.

PONE-D-24-14821R2

Dear Dr. Larsen,

We’re pleased to inform you that your manuscript has been judged scientifically suitable for publication and will be formally accepted for publication once it meets all outstanding technical requirements.

Kind regards,

Maria Lina Tornesello

Academic Editor

PLOS ONE
---

## [Editor Report · Acceptance letter]

PONE-D-24-14821R2

PLOS ONE

Dear Dr. Larsen,

I'm pleased to inform you that your manuscript has been deemed suitable for publication in PLOS ONE. Congratulations! Your manuscript is now being handed over to our production team.

Kind regards,

on behalf of

Dr. Maria Lina Tornesello

Academic Editor

PLOS ONE